# Do hospitalisations for ambulatory care sensitive conditions reflect low access to primary care? An observational cohort study of primary care usage prior to hospitalisation

Sabine I Vuik,[1] Gianluca Fontana,[1] Erik Mayer,[2] Ara Darzi[1,2]

[1]Institute of Global Health Innovation, Imperial College, St Mary's Hospital, London, UK
[2]Department of Surgery, Imperial College, St Mary's Hospital, London, UK

**Correspondence to**
Sabine I Vuik;
s.vuik@imperial.ac.uk

## ABSTRACT

**Objectives** To explore whether hospitalisations for ambulatory care sensitive conditions (ACSCs) are associated with low access to primary care.

**Design** Observational cohort study over 2008 to 2012 using the Clinical Practice Research Datalink (CPRD) and Hospital Episode Statistics (HES) databases.

**Setting** English primary and secondary care.

**Participants** A random sample of 300 000 patients.

**Main outcome measures** Emergency hospitalisation for an ACSC.

**Results** Over the long term, patients with ACSC hospitalisations had on average 2.33 (2.17 to 2.49) more general practice contacts per 6 months than patients with similar conditions who did not require hospitalisation. When accounting for the number of diagnosed ACSCs, age, gender and GP practice through a nested case–control method, the difference was smaller (0.64 contacts), but still significant (p<0.001). In the short-term analysis, measured over the 6 months prior to hospitalisation, patients used more GP services than on average over the 5 years. Cases had significantly (p<0.001) more primary care contacts in the 6 months before ACSC hospitalisations (7.12, 95% CI 6.95 to 7.30) than their controls during the same 6 months (5.57, 95% CI 5.43 to 5.72). The use of GP services increased closer to the time of hospitalisation, with a peak of 1.79 (1.74 to 1.83) contacts in the last 30 days before hospitalisation.

**Conclusions** This study found no evidence to support the hypothesis that low access to primary care is the main driver of ACSC hospitalisations. Other causes should also be explored to understand how to use ACSC admission rates as quality metrics, and to develop the appropriate interventions.

## BACKGROUND

Chronic ambulatory care sensitive conditions (ACSCs) are conditions like diabetes, chronic obstructive pulmonary disease and angina that are cared for in the community by primary care providers.[1–3] With high-quality primary care—which is timely,[4 5] effective,[4] continuous[5–8] and accessible[1 9–15]—the need

### Strengths and limitations of this study

► This study uses linked primary and secondary care data to explore whether there exists a patient-level, temporal relation between low usage of primary care and ambulatory care sensitive condition (ACSC) hospitalisations.

► It considers both long-term (5 years) and short-term (6 months prior to hospitalisations) primary care usage.

► To account for confounders, this study uses a nested case–control design, in addition to looking at average rates in the population.

► While primary care usage can be considered a proxy of 'realised' access to care, it remains only a proxy and does not consider unmet needs.

► This study only looks at access to primary care, and does not explore other quality aspects of primary care that may be related to ACSC hospitalisation, such as timeliness and effectiveness.

for hospital care for these conditions can be reduced. Therefore, high rates of emergency hospitalisations for chronic ACSCs can be an indication of lower quality primary care.[1 16 17]

In addition to providing a measure of low care quality, emergency hospitalisations for ACSC also reflect a major health system expenditure,[18] and a negative patient experience—covering all elements of the Triple Aim.[19] As a result, the rate of ACSC hospitalisation has been widely used to measure the overall performance of primary care, by national organisations in the UK, the USA, Canada and Australia.[20–23]

Despite the widespread use of these metrics, empirical evidence of their validity is limited.[24] In particular, ACSC hospitalisations are often considered a reflection of access to primary care,[1 9–15 25] but the actual relation between primary care access and hospitalisations for ACSCs remains unclear.[24 26] Various studies have

found low access to primary care physicians to be related to higher rates of ACSC hospitalisation.[9–13 15 27] However, these are ecological studies using high-level proxies for primary care access, such as the density of primary care physicians in a certain region.[9–11 27] Other studies have found no relation between primary care access and ACSC hospitalisation rates.[5 28 29]

Understanding what drives hospitalisations for ACSCs is the first step in developing effective policies to address the issue. This paper aims to identify whether there is a relation between low access to primary care and ACSC hospitalisations. Using administrative data linked across primary and secondary care, it explores this relation at a patient level. It considers both long-term access to primary care, reflecting the long-term management of the condition, and access in the 6 months prior to a hospitalisation.

## METHODS
### Study overview
Using a retrospective cohort study, this paper analyses the relation between primary care usage and hospitalisations for ACSCs at the patient level. In addition to exploring population-level rates, matched case–control samples were selected and compared to adjust for a range of confounders. The study looks at both long-term usage, as measured over 5 years, and usage directly prior to ACSC hospitalisation, as measured over the 6 months preceding hospitalisation.

To understand the long-term levels of primary care usage for patients with ACSC hospitalisations, the average number of GP contacts per 6 months was calculated from 5 years of data. To contextualise this usage rate and determine whether it is low or high, it was compared with the usage rates of two reference populations: patients without any diagnosed ACSCs, and patients with diagnosed ACSCs but no qualifying emergency hospitalisations. (Note that, for simplicity, the second group is called 'patients with ACSCs but no hospitalisations'. This only refers to qualifying emergency admissions for ACSCs as defined below. Patients in this group may still have had elective hospitalisations or hospitalisations for non-ACSC causes.)

To explore whether a temporary lack of primary care contributes to ACSC hospitalisations, primary care usage in the 6 months directly prior to the hospitalisation was also calculated. In addition, usage rates for six consecutive 30-day intervals were calculated to study changes in care access leading up to an ACSC hospitalisation. These retrospective analyses were done at the ACSC hospitalisation level rather than the patient level by identifying the 6 months of GP usage prior to each individual ACSC hospitalisation. All hospitalisations were included, even if they occurred in the same person, or if their 6 months of prior activity overlapped with another ACSC hospitalisation for that patient.

### Data sources and study sample
Using the Clinical Practice Research Datalink (CPRD) and Hospital Episode Statistics (HES), a database containing a random sample of 300 000 English patients and their 5-year (2008–2012) primary care and secondary care usage was created. Patients were eligible for inclusion in the study if they were registered with a CPRD participating GP throughout the entire observation period, and if their primary care data could be linked to their secondary care data in HES. Linkage of the data was done by CPRD, and required consent and the availability of matching National Health Service (NHS) number and other personal details (note: this information was not available to the authors, who only had anonymous data). The sample size was set at 300 000, which is the maximum sample size CPRD ordinarily allows.

### Main outcome measures
The main measures in this study were hospitalisations for ACSCs, and primary care usage.

### Hospitalisations for ACSC admissions
In the HES data, ACSC hospitalisations were identified following the definitions below. To ensure 6 months of primary care activity was available preceding each ACSC hospitalisation, only hospitalisations in the final 4.5 years were included. All ACSC hospitalisations were included, even if they were for the same person or within the 6-month period prior to another ACSC hospitalisation.

The definition of ACSCs generally includes both chronic and acute conditions.[30] This study focuses on chronic ACSCs only, as it is interested in long-term disease management in the primary care setting. In addition, it allows a comparator group to be created consisting of people with similar chronic conditions but no hospitalisations. A wide variety of ACSC conditions and coding practices exists.[30] This research defines ACSCs according to the International Statistical Classification of Diseases and Related Health Problems, 10th revision (ICD-10) codes used in the NHS England Outcomes Framework, as these are specific to English administrative data.[20]

The diagnosis codes for ACSC hospitalisations were also used to identify patients who have been diagnosed with those conditions in the 300 000 patients sample. Patients' morbidity profiles were based on a combination of both the HES and CPRD dataset. Without data outside the 5-year study period, we were unable to identify whether a diagnosis code at a certain time point is a newly onset disease, or reflects treatment for an existing condition. Therefore, no temporal analysis was possible, and a diagnosis at any time point was treated as a positive flag for that condition. To be able to find ACSC diagnoses in the CPRD database, the ICD-10 definitions had to be translated to READ codes, which are used in primary care. A mapping created by the NHS Health and Social Care Information Centre was used for this purpose.[31]

**Table 1** Sample characteristics for the three population groups

| | People with no ACSCs | People with ACSCs but no hospitalisations | People with hospitalisations for ACSCs | Total population |
|---|---|---|---|---|
| N | 229 631 | 65 179 | 5190 | 300 000 |
| Age (average) | 41 | 58 | 63 | 45 |
| Gender | | | | |
| Female (%) | 50 | 53 | 50 | 51 |
| Male (%) | 50 | 47 | 50 | 49 |
| No of diagnosed ACSCs (average) | 0 | 1.47 | 2.60 | 0.36 |
| No of hospitalisations for ACSCs over 4.5 years (average) | 0 | 0 | 1.44 | 0.02 |

ACSC, ambulatory care sensitive condition.

## Care usage

This study aims to measure access to care. While it is difficult to measure 'potential' care access, which includes unmet needs, 'realised' care access can be measured in terms of care usage rates.[32] Therefore, usage rates are used as a proxy for access. It is important to note that this metric measures the patient's frequency of contact with healthcare providers, but it does not consider other aspects of access such as the length of the contact or instances where a patient is unable to get an appointment.

Usage rates were measured for primary care by counting the number of GP contacts, including surgery and clinic attendances, home visits and telephone calls. No distinction was made between different types of practice staff, and care could have been provided by practice nurses and other qualified staff. Therefore, a 'GP contact' should be interpreted as a contact with the GP practice rather than a specific GP.

## Statistical analysis

Using SPSS Statistics V.23,[33] descriptive statistics, including average usage rates, were calculated for the overall population and the three population groups. Independent t-tests were used to compare the usage rates in the population with ACSCs hospitalisations to the population with ACSCs but no hospitalisations.

To adjust for timing differences as well as confounders such as age and ACSC counts, a nested case–control method was used. Using the FUZZY extension in SPSS, patients with an ACSC hospitalisation were matched to a control, based on gender, number of diagnosed ACSCs, GP practice (to account for provider-level differences) and 10-year age bands. The long-term primary care usage rates were compared using independent t-tests between the cases and controls. For the usage prior to ACSC hospitalisation, the same 6-month time period preceding the hospitalisation of the cases was used for the controls, to account for seasonal or other timing differences.

To analyse the change in usage rate leading up to an ACSC hospitalisation, a general linear model for repeated measures was used with the usage rates for the six 30-day intervals as the response variable. These intervals were compared pairwise as well, adjusting the significance level for multiple tests using a Bonferroni correction.

## RESULTS

In the database, 7467 hospitalisations for ACSCs were identified for the sample of 300 000 people over 4.5 years. This equates to an admission rate of 553 per 100 000 people per year, which is in line with crude ACSC admission rates for regions in England in 2014/15, which range from 137 to 1384 per 100 000.[34]

Nearly one-fourth (70 369) of people in the population had diagnoses for one or more ACSCs (see table 1). Of these people, 7% (5190) had one or more hospitalisations for their ACSCs, with 4017 people having one ACSC hospitalisation during the 4.5 years and 1173 people more than one.

For 76% (3957) of all people with an ACSC hospitalisation an exactly matched control was found (see table 2). This subgroup accounted for 70% of all ACSC hospitalisations in the database. While a 10-year age band was

**Table 2** Sample characteristics of the nested cases and controls

| | Cases | Controls |
|---|---|---|
| N | 3957 | 3957 |
| Age (average) | 62.02 | 62.27 |
| Gender | | |
| Female (%) | 50 | 50 |
| Male (%) | 50 | 50 |
| No of diagnosed ACSCs (average) | 2.13 | 2.13 |

ACSC, ambulatory care sensitive condition.

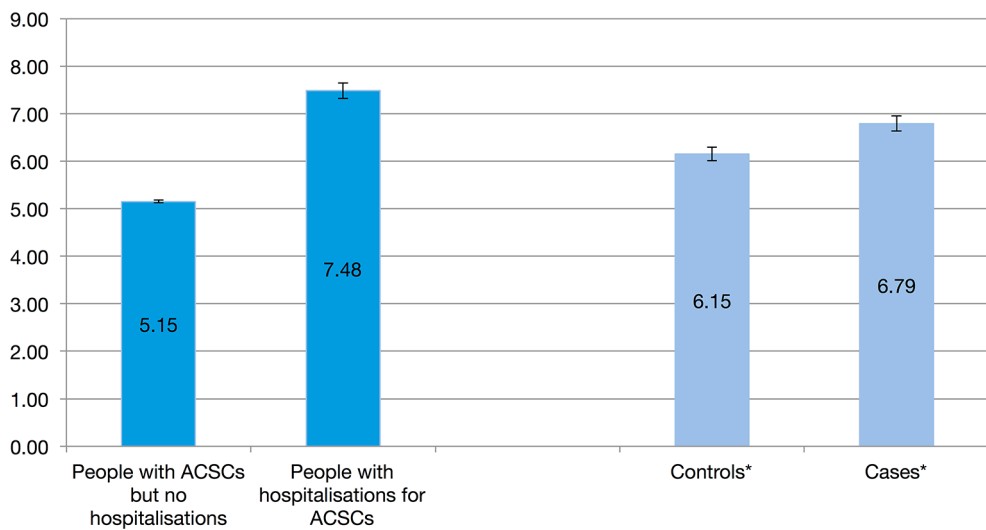

*\* Samples matched on ten-year age band, number of diagnosed ACSCs, GP practice and gender*

⊺ *95% CI*

**Figure 1** Average no of general practice (GP) contacts per 6 months. ACSC, ambulatory care sensitive condition.

used to match people rather than exact age, there was no significant (p=0.62) age difference between the two groups.

Over the full study period of 5 years, patients with ACSC hospitalisations had significantly more (2.33, p<0.001) GP contacts per 6 months (7.48, 95% CI 7.32 to 7.64) than people with similar conditions who did not require hospitalisation (5.15, 95% CI 5.12 to 5.18) (see figure 1). The nested case–control method adjusts for the number of diagnosed ACSCs and other confounders. While the difference between these two groups was smaller, people with ACSC hospitalisations still had significantly more (0.64, p<0.001) GP contacts than their controls.

Since the usage directly prior to a hospitalisation is measured over only 6 months, there may be seasonal differences for this metric. Therefore, primary care usage prior to hospitalisation in the cases was compared with usage in the controls over the same 6-month time period. People with ACSC hospitalisations had significantly more (1.55, p<0.001) primary care contacts in the 6 months before hospitalisations (7.12, 95% CI 6.95 to 7.30) than their controls during the same 6 months (5.57, 95% CI 5.43 to 5.72).

To explore differences in primary care use leading up to ACSC hospitalisations, the usage rates of six consecutive 30-day intervals were calculated (see figure 2). GP

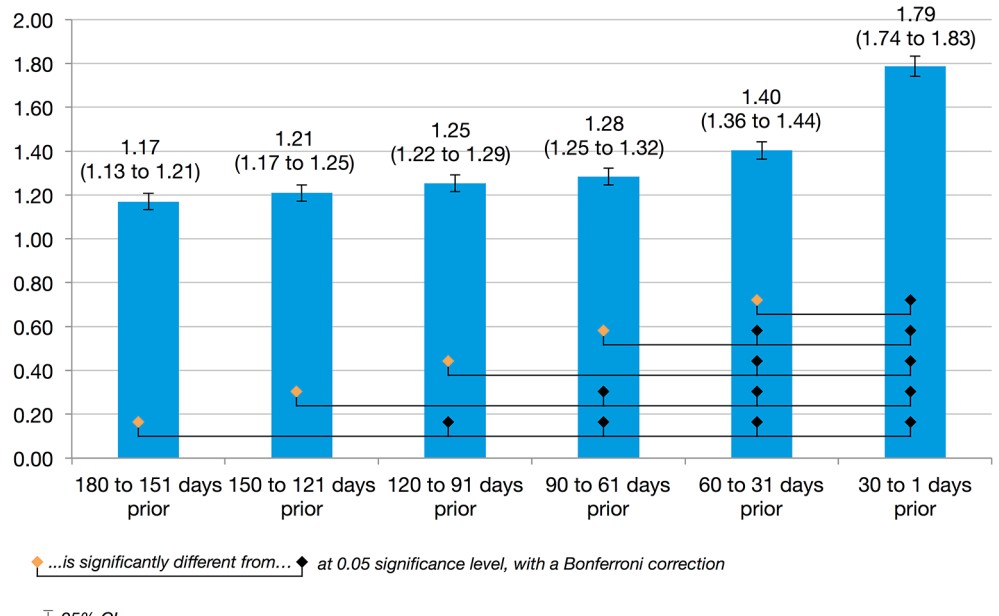

◆ *...is significantly different from...* ◆ *at 0.05 significance level, with a Bonferroni correction*

⊺ *95% CI*

**Figure 2** Average no of general practice (GP) contacts prior to an ambulatory care sensitive condition (ACSC) hospitalisation.

contacts were skewed towards the months closer to the hospitalisation date. A repeated measures general linear model showed that the mean number of GP contacts is significantly different ($p<0.001$) across the six 30-day intervals. Pairwise comparisons show that the peak in GP contacts in the final 30 days prior to hospitalisation is significantly different from each of the five preceding months. (Note: the sum of these intervals (8.11) is slightly less than the 6-month usage (8.19), as the latter is calculated over 182 days rather than 180 to reflect six calendar months, allowing comparison to the 5-year usage rates).

## DISCUSSION

The primary objective of this study was to explore whether low access to primary care is related to hospitalisations for ACSCs. No evidence was found of low primary care usage either over the long term, which could affect the ongoing management of ACSCs, or in the short- term, prior to an ACSC hospitalisation. The results show that, over 5 years, patients with ACSC hospitalisations in fact use more GP services compared with other patients with the same number of diagnosed ACSCs, age, gender and GP practice.

Neither does there appear to be a temporary drop in usage. In the 6 months immediately prior to an ACSC hospitalisation, usage is higher than the 5-year average. People with ACSC hospitalisations use more primary care than matched controls over the same 6-month period. In addition, the average usage rate increased every month nearer to the hospitalisation, with a significant peak in the last 30 days prior to hospitalisation. Overall, this research finds no evidence that low access to GP services is a key driver of ACSC hospitalisation.

This study found a statistically significant relation between primary care usage and ACSC hospitalisations, where the usage rate increases prior to hospitalisation. Future research should explore whether this correlation can be used to improve risk models. Predictive models could be developed that predict ACSC hospitalisations based on a change in GP usage, allowing preventive interventions to be delivered.

An important strength of this study is the use of linked, administrative data at the patient level. Rather than using population-level proxies for access, as has been done in previous studies on this topic,[9–11 27] this study linked primary care usage and ACSC hospitalisations at the patient level. In addition, it considered both long-term care use and care usage immediately preceding an ACSC hospitalisation. Using this approach, the direct, temporal relation between a patient's access to primary care and ACSC hospitalisation could be explored.

The research has a number of limitations, in addition to the well-documented general limitations associated with using administrative healthcare data.[35] First, while usage approximates access to care at the patient level, it is still only a proxy. No differentiation can be made between low usage due to limited access to services, and other causes such as patients not seeking care. In addition, this proxy does not take into account any unmet needs. Usage rates for patients with ACSC hospitalisations were in line with those of people with similar disease profiles, but their actual need for care may have been even higher—as suggested in their eventual hospitalisation. The fact that ACSC hospitalisations are not associated with lower access to primary care does not preclude that more primary care could prevent them.

This is a crucial limitation of this research, and of any similar analyses based on administrative data. To fully understand what drives ACSC hospitalisations at the individual patient level, case note reviews or interviews are needed. Administrative data only cover actualised care. A case note review study could verify the results from this research, as well as the overall approach of using administrative data to measure care needs.

Second, this paper only explores the relation between low access to primary care and ACSC hospitalisations. ACSC hospitalisation rates are used as a high-level measure of primary care, and can been linked to many other aspects of care quality, such as timeliness,[4 5] effectiveness,[4] continuity,[5–7] safety and appropriateness.[36] Further research is needed to understand the specific and relative impact of these factors on ACSC hospitalisations.

Third, there remains the possibility of confounding. While a nested case–control method was used to account for diagnosed ACSCs, age, gender and provider differences, other factors such as urban/rural environments, ethnicity[25] and socioeconomic status[5 37] could be influencing the results. While matching on GP practice partially mitigates these issues due to patients living in the same geographic area, some degree of confounding remains possible.

In addition, cases were matched based on the number of ACSCs rather than specific conditions—limiting the comparability of the cases and controls. This approach was chosen to ensure more matches could be found, and to create a quantitative measure of health status. Nevertheless, overall health remains an important potential confounder, particularly due to its direct impact on care usage. This is another area where case reviews would provide a level of detail that is not available in administrative data.

The cases for which a matched control was found had slightly less primary care activity and ACSC hospitalisations than the overall population with ACSC hospitalisations. This could be due to a lack of matched controls for extreme cases with a very high age or ACSCs count, who could also be expected to use more care. The exclusion of extreme cases is a limitation of the methods used in this study, and there is an opportunity for future research to explore what drives ACSC hospitalisations in this high-needs group.

The results presented here have important implications for clinicians and policymakers. This study provides no evidence that, as is commonly believed, low access to primary care is a leading cause of ACSC hospitalisations.

This would mean that ACSC hospitalisation rates, while informative, should not be used specifically as indicators of access to primary care.

More research is needed into new models of care to meet the needs of patients with ACSC hospitalisations. Policymakers and clinicians trying to reduce emergency hospitalisation should consider policies that focus on a variety of aspects of primary care, including effectiveness and continuity of care, rather than focusing on access alone. Tailored interventions aimed at high-risk patients should also be explored, including remote support and observational beds.

## CONCLUSION

This research found no patient-level, temporal relation between low usage of primary care services and hospitalisations for ACSCs. While there may be other aspects of primary care quality that impact ACSC hospitalisation rates, there is no evidence that low access to care is a key driver. More research is needed into the actual causes of ACCS hospitalisations, across care settings, to allow accurate interpretation of ACSC hospitalisation rates and to develop effective interventions.

**Contributors** SV designed the study, created the database, analysed the data, and drafted and revised the paper. She is the guarantor. GF and EM contributed to the design of the study, analysed the results and revised the draft paper. AD contributed to the design of the study and revised the draft paper. All have approved the final version for publication.

**Funding** The research was funded by the National Institute for Health Research (NIHR) Biomedical Research Centre based at Imperial College Healthcare NHS Trust and Imperial College London. The views expressed are those of the authors and not necessarily those of the NHS, the NIHR or the Department of Health. This project was also supported by the Peter Sowerby Foundation.

**Disclaimer** This study is based on data from the Clinical Practice Research Datalink obtained under license from the UK Medicines and Healthcare Products Regulatory Agency. However, the interpretation and conclusions contained in the study are those of the authors alone. The study was approved under reference number 14_211RA.

**Competing interests** None declared.

**Provenance and peer review** Not commissioned; externally peer reviewed.

**Data sharing statement** The SPSS syntax documents detailing the steps taken to create the analysis databases from raw CPRD and HES data are available upon request from the corresponding author.

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
