## [Reviewer comments · BMJ Open]

ARTICLE DETAILS

TITLE (PROVISIONAL)	Do hospitalisations for Ambulatory Care Sensitive Conditions reflect low access to primary care? An observational cohort study of primary care utilisation prior to hospitalisation
AUTHORS	Vuik, Sabine; Fontana, Gianluca; Mayer, Erik; Darzi, Ara

VERSION 1 - REVIEW

REVIEWER	Leif I Solberg, MD HealthPartners Institute, USA
REVIEW RETURNED	11-Jan-2017

GENERAL COMMENTS	I was glad to see this attempt to validate one of the key assumptions underlying the ACSC list of conditions created by expert opinion based on the belief that these hospital admissions often/usually reflect inadequate ambulatory care. One aspect of such inadequate care is presumed to be limitations in accessing that care. The main limitation in the methods used in this study to test this assumption is that reliance on large data bases to compare the frequency of contacts is a big step removed from actually reviewing the cases themselves. In other words, demonstrating that the number of GP services for these patients was similar to those for patients without ACSCs does suggest the possibility that access was the same, but doesn't prove it. If one reviewed the actual case histories, it is possible that the patients with ACSCs actually needed or wanted even more access, but were unable to obtain it. You hint at this problem in the Discussion section on limitations, but don't address it as directly and clearly as I think you should. As a result, the conclusion in both the narrative and abstract seems to me to somewhat overstate the extent to which you have proven that access was not a problem. Please temper the conclusion statements and more explicitly suggest the need for additional research that obtains either patient reports or record evidence that access to care was no different for these patients than those without ACSCs. Otherwise, I think the methods used and writing style are good and I could not identify any other problems that need revision.
--

REVIEWER	William B Weeks, MD, PhD, MBA The Dartmouth Institute for Health Policy and Clinical Practice, USA
REVIEW RETURNED	23-Jan-2017

GENERAL COMMENTS	This is a well-written, informative, and important paper that explores actual utilization of primary care services prior to admission for chronic ACSC....the paper also uses matched controls to address the reality that those with ACSC hospitalizations had more ACSCs
--

	and were older than those in the general population with ACSCs but not hospitalizations. The statistical methods are thorough, though someone with more expertise than me should review the paper to ensure they are used correctly. While the paper does find a statistically different number of GP visits prior to ACSC admission (with # of visits being higher in the admissions group), the differences are substantially lower when using the matched controls. Further, the number of emergency non ACSC admissions are 40% higher in the cases than the controls. Together, with the accelerating use of GP visits in the months preceding ACSC admission, this indicates to me that a patients' health status is becoming more out-of-control. So while it is perhaps true that primary care access as reflected in visits realized might not be the issue, what the authors were unable to determine is whether primary care availability met primary care demand. And while the authors state that more needs to be done here, it does seem that they might be a bit more bold and state that 1. their work suggests that predictive modeling might be helpful in identifying patients at risk for ACSC admission ('velocity' of use changing over time) and 2. that new models of care that meet increasing patient demands for care fairly immediately before admissions that really should be preventable (i.e., even more primary care visits? more access to specialists? more use of observation beds? more remote support?) should be tested. Thanks for the opportunity to review this really nicely done paper.
--	--

REVIEWER	John Busby Queens University Belfast, Northern Ireland
REVIEW RETURNED	13-Feb-2017

GENERAL COMMENTS	Overview This paper uses routinely collected administrative data from primary and secondary care to investigate the association between primary care access and emergency admissions for ambulatory care sensitive conditions. This is clearly a very important issue and particularly timely given the overwhelming demand being experienced within A&E departments across the UK. ACSC admissions make up around a sixth of all emergency admissions, so improved care for these patients could be of significant importance to policymakers. Overall, I found this paper slightly difficult to follow and quite muddled in places. In particular, I did not see the relevance of investigating the impact of primary care access on elective admissions, outpatient attendances etc., and the analysis exploring how primary care utilisation changes in the lead up to admission (given these are not mentioned in the aims). My main concern with the paper is the choice of exposure variable (primary care utilisation) as a proxy for primary care access. This could be affected by several other factors including underlying health, rurality, ethnicity etc. which, at the very least will manifest in substantial measurement error and regression dilution bias (i.e. an attenuation of the estimated effect towards the null). However, if some of these factors are also related to the need for hospitalisation, which I think is likely
--

(particularly due to underlying health), this will lead to confounding which could undermine the validity of the entire study. I remain unconvinced that the strategies employed by the authors to adjust for this potential confounding are sufficiently robust to have full confidence in these findings.

Abstract

- The main outcome measure is listed as primary care access – shouldn't this be ACSC hospitalisations?
- I don't think it is particularly clear that primary care access is being measured in the time prior to the hospital admission

Introduction

- The authors should mention that some ACSCs, such as influenza, are potentially avoidable through better prevention
- I find it surprising that the authors did not reference the Huntly systematic review as it seems highly relevant (Which features of primary care affect unscheduled secondary care use? A systematic review, *BMJ Open*, 2014). They may also wish to reference some recent work I have been involved in exploring the causes of ACSC admissions (A systematic review of the magnitude and cause of geographic variation in unplanned hospital admission rates and length of stay for ambulatory care sensitive conditions, *BMC HSR*, 2015)
- I don't think the mechanism through which improved access can lead to reduced hospitalisations is adequately described. In general, I think a more detailed description of what ACSCs are, and how admission for these conditions might be avoided through improved primary care, would be useful for the uninitiated.
- I think it is obvious that other non-primary care factors can effect ACSC hospitalisation – this point is somewhat laboured.

Methods

- Most of the cohort is patients who do not have any ACSC. I think any unadjusted comparison between this group and the ACSC group (such as in figure 1) is fairly meaningless. They are clearly very different groups in terms of age etc.
- I didn't follow the rationale for the sample size. Surely it's a case of 'the more the better' – there was no need to restrict this sample to that of a largish CCG (300,000 patients) if more data were available.
- I agree that measuring potential access to care is difficult. However I have serious concerns around the proxy measure used by the authors due to potential confounding by underlying health, rurality and a host of other factors that could feasibly be related to primary care utilisation and risk of hospital admission.
- The authors have attempted to adjust for this in one analysis by using matched controls, however I remain unconvinced by their matching strategy. Firstly, matching on the number of ACSCs may not be appropriate as some ACSC may impact the risk of admission much more than others (e.g. equating asthma with diabetes does not seem right). Secondly, I do not understand why they restricted their matching to only ACSC conditions. Other non-ACSC conditions are likely to influence the risk of admission (e.g. previous stroke) and could well differ between the hospitalised and non-hospitalised groups. Overall, I think the risk of bias in this matched case-control analysis is high and so its results cannot be fully trusted.
- I don't really see the relevance of the analysis exploring the impact of impact of primary care access on elective admissions, outpatient attendances etc., and the analysis exploring how primary care utilisation changes in the lead up to admission (given these are not mentioned in the aims)

	Results  • An estimate of effect size (i.e. difference between cases and controls) should be given in the text and the results tables, currently only p-values are provided. • The results for the patients with no ACSC are given in Figure 1 but not mentioned in the text. Really these patients offer very little to the study and could safely be removed from the paper entirely. • The paper states that 'As observed for the long-term utilisation, there are factors such as the number of ACSCs that influence primary care utilisation'. I cannot see the data to support this statement. Can the authors clarify? Discussion  • I don't follow the rationale that because patients with an ACSC hospital admission were also more likely to have an outpatient procedure, this could have an impact on intervention design. Can the authors clarify? Are they saying that patients are more likely to be admitted as an emergency because they have had more outpatient appointments? This seems unlikely. • The authors state that 'this paper only explores the relation between low access to primary care and ACSC hospitalisation'. This does not appear to be true – for example they also examine outpatient utilisation • The authors do acknowledge that confounding could drive their results however I think, given the limitations of the study, this should be stronger and explicitly mention that underlying health could be an important confounder which is not fully addressed in their case-control study. • I think that the conclusions stating that policymakers should focus on other aspects of primary care (e.g. continuity) are a little strong given the limitations of the study.
--	---

REVIEWER	Miguel Martín Facultad de Medicina Unidad de Bioestadística Universidad Autónoma de Barcelona
REVIEW RETURNED	28-Mar-2017

GENERAL COMMENTS	Although the described design is defined as a control case the analysis of the effect is performed by a linear model of repeated measures that does not clearly state what the variable response is. There are several aspects that I do not see clearly can facilitate the conclusion of the story. Is the number of GP visits a measure of accessibility or severity of cases? Is accessibility or resolving capability being measured? Why not carry out a follow-up analysis of the Poisson type, or related, with adjustment of the extravariance due to the appearance of the ACSC hospitalization interdependence? It is another approach but it provides the assessment of the risk of being hospitalized for the previous fact of "accessibility" and not so much the rate of hospitalization
--

REVIEWER	Gisele M Carriere, Senior Social Science Researcher Health Analysis Division, Statistics Canada, Canada
-----------------	---

REVIEW RETURNED

19-Apr-2017

GENERAL COMMENTS

This is an important research question. A strength is the use of large population sample, especially since creation of nested case controls using exact matching of characteristics can challenge large enough sample sizes to look at relatively rare events like ACSC hospitalizations.

I answered 'no' to sufficient method description because the following things require some clarification: 1) how was the 'index' (my label) hospitalization selected for retrospective analysis of GP contacts? Very likely people with ACSCs had more than one hospitalization in this 5 year frame or even within 6 month frames. Perhaps several hospital events were chosen and the months before for each were compiled and pooled? Or maybe you removed people once they had a hospitalization within a given 6 month frame and considered them only once in that frame? 2) Similarly, people may have 'accumulated' numbers of diagnosed ACSCs as time went on across the 5 years or even within a 6 month interval. How did you assign them one given # of ACSCs? What base year or month would have been used? I understand that 'time until hospitalization' was not of interest here (where a proportional hazards model would be used instead) -- rather this is a logistic approach. But I'm wondering how you considered changes that may have occurred for people to the #s of ACSCs over intervals used in the linear model.? It might be better to change language from "To explore changes in primary care " (implying trend or survival analysis) to 'To explore differences in primary care use leading up to ACSC hospitalizations...'. Next RE: Linkage - what method, or linkage approach was used to link primary care to secondary care in HES? What linkage rates resulted? Could you describe characteristics of those who could not be linked? i.e. perhaps they did not have GPs or other primary care providers? Is this a study of 'low access' (defined as ?) or does this examine association between primary and secondary care in general? Minor: you have a typo in the last paragraph before the Discussion - 'significant different' should read 'significantly different'. Lastly, Results are consistent to results by the following (2011) that showed individuals with ACSC hospitalizations were more likely to be users of primary and specialist care services and to have a regular doctor; risk factors for ACSC hospitalizations include such characteristics as having co-morbidities, low income and lifestyle some of which may be addressed by primary care however where other solutions to meet these health care needs my reside outside of the scope of health care: Sanmartin C, Khan S, Longitudinal Health Administrative Data Research team. Hospitalizations for ambulatory care sensitive conditions (ACSC): The factors that matter. Health Research Working Paper Series (Catalogue 82-622-X, No. 007) Ottawa: Statistics Canada, 2011. Thank you for the opportunity to review your research.

VERSION 1 – AUTHOR RESPONSE

Reviewer: 1

Reviewer Name: Leif I Solberg, MD

Institution and Country: HealthPartners Institute, USA

Please state any competing interests or state 'None declared': None

Please leave your comments for the authors below

I was glad to see this attempt to validate one of the key assumptions underlying the ACSC list of conditions created by expert opinion based on the belief that these hospital admissions often/usually reflect inadequate ambulatory care. One aspect of such inadequate care is presumed to be limitations in accessing that care. The main limitation in the methods used in this study to test this assumption is that reliance on large data bases to compare the frequency of contacts is a big step removed from actually reviewing the cases themselves. In other words, demonstrating that the number of GP services for these patients was similar to those for patients without ACSCs does suggest the possibility that access was the same, but doesn't prove it. If one reviewed the actual case histories, it is possible that the patients with ACSCs actually needed or wanted even more access, but were unable to obtain it. You hint at this problem in the Discussion section on limitations, but don't address it as directly and clearly as I think you should. As a result, the conclusion in both the narrative and abstract seems to me to somewhat overstate the extent to which you have proven that access was not a problem. Please temper the conclusion statements and more explicitly suggest the need for additional research that obtains either patient reports or record evidence that access to care was no different for these patients than those without ACSCs.

Otherwise, I think the methods used and writing style are good and I could not identify any other problems that need revision.

***We fully agree with the limitations mentioned by Dr Solberg, and have edited the paper to make them more explicit, and tempered the conclusion statement and abstract. The suggestion of using case reviews instead of administrative data has been added in two places in the discussion.

Reviewer: 2

Reviewer Name: William B Weeks, MD, PhD, MBA

Institution and Country: The Dartmouth Institute for Health Policy and Clinical Practice, USA

Please state any competing interests or state 'None declared': none

Please leave your comments for the authors below

This is a well-written, informative, and important paper that explores actual utilization of primary care services prior to admission for chronic ACSC....the paper also uses matched controls to address the reality that those with ACSC hospitalizations had more ACSCs and were older than those in the general population with ACSCs but not hospitalizations.

The statistical methods are thorough, though someone with more expertise than me should review the paper to ensure they are used correctly.

While the paper does find a statistically different number of GP visits prior to ACSC admission (with # of visits being higher in the admissions group), the differences are substantially lower when using the matched controls. Further, the number of emergency non ACSC admissions are 40% higher in the cases than the controls. Together, with the accelerating use of GP visits in the months preceding ACSC admission, this indicates to me that a patients' health status is becoming more out-of-control. So while it is perhaps true that primary care access as reflected in visits realized might not be the issue, what the authors were unable to determine is whether primary care availability met primary care demand. And while the authors state that more needs to be done here, it does seem that they might be a bit more bold and state that 1. their work suggests that predictive modeling might be helpful in identifying patients at risk for ACSC admission ('velocity' of use changing over time) and 2. that new models of care that meet increasing patient demands for care fairly immediately before

admissions that really should be preventable (i.e., even more primary care visits? more access to specialists? more use of observation beds? more remote support?) should be tested.

Thanks for the opportunity to review this really nicely done paper.

***We'd like to thank Dr Weeks for his feedback, and have incorporated both suggestions in the discussion.

Reviewer: 3

Reviewer Name: John Busby

Institution and Country: Queens University Belfast, Northern Ireland

Please state any competing interests or state 'None declared': None declared

Please leave your comments for the authors below

Overview

This paper uses routinely collected administrative data from primary and secondary care to investigate the association between primary care access and emergency admissions for ambulatory care sensitive conditions. This is clearly a very important issue and particularly timely given the overwhelming demand being experienced within A&E departments across the UK. ACSC admissions make up around a sixth of all emergency admissions, so improved care for these patients could be of significant importance to policymakers.

Overall, I found this paper slightly difficult to follow and quite muddled in places. In particular, I did not see the relevance of investigating the impact of primary care access on elective admissions, outpatient attendances etc., and the analysis exploring how primary care utilisation changes in the lead up to admission (given these are not mentioned in the aims).

***We appreciate Dr Busby's confusion around the use of other settings in our paper, and have decided to remove this part to make the paper and its argument clearer.

The primary care changes in the lead up to admission were included to explore the short-term impact of primary care use (vs. long-term, which is measured over five years). We have clarified this in the paper.

My main concern with the paper is the choice of exposure variable (primary care utilisation) as a proxy for primary care access. This could be affected by several other factors including underlying health, rurality, ethnicity etc. which, at the very least will manifest in substantial measurement error and regression dilution bias (i.e. an attenuation of the estimated effect towards the null). However, if some of these factors are also related to the need for hospitalisation, which I think is likely (particularly due to underlying health), this will lead to confounding which could undermine the validity of the entire study. I remain unconvinced that the strategies employed by the authors to adjust for this potential confounding are sufficiently robust to have full confidence in these findings.

***We fully acknowledge Dr Busby's concerns on confounding, this is a crucial issue and something that we have spent a lot of time on. We had little to no data available on factors such as rurality and ethnicity, making it difficult to directly account for these important confounders. We tried to mitigate this issue by matching our cases and controls on GP practice. It is by no means perfect, but it makes it more likely that the cases and controls are of a similar rural/urban setting and from the same environment. Underlying health we approximated using ACSC count as a matching variable, but again we fully agree that this is not a perfect indication of overall health status. We have added this explanation to the discussion, to ensure the findings are set in the right context.

Abstract

- The main outcome measure is listed as primary care access – shouldn't this be ACSC hospitalisations?

***Thank you, this has been changed.

- I don't think it is particularly clear that primary care access is being measured in the time prior to the hospital admission

***It is being measured prior to admission for the short-term analysis, but in the abstract the first half of the results talks about the long-term utilisation that is measured over the full five years. We've clarified this.

Introduction

- The authors should mention that some ACSCs, such as influenza, are potentially avoidable through better prevention

***Thank you for highlighting this. Indeed acute ACSCs such as influenza and UTI have a strong preventive component that also influences hospitalisation rates, which is why for our analysis we focused only on chronic ACSCs. We've now clarified this in the introduction.

- I find it surprising that the authors did not reference the Huntly systematic review as it seems highly relevant (Which features of primary care affect unscheduled secondary care use? A systematic review, BMJ Open, 2014). They may also wish to reference some recent work I have been involved in exploring the causes of ACSC admissions (A systematic review of the magnitude and cause of geographic variation in unplanned hospital admission rates and length of stay for ambulatory care sensitive conditions, BMC HSR, 2015)

***Thank you for bringing these two interesting studies to our attention. We've included them in our paper.

- I don't think the mechanism through which improved access can lead to reduced hospitalisations is adequately described. In general, I think a more detailed description of what ACSCs are, and how admission for these conditions might be avoided through improved primary care, would be useful for the uninitiated.

***We agree and have expanded the introduction to provide more information on ACSCs

- I think it is obvious that other non-primary care factors can effect ACSC hospitalisation – this point is somewhat laboured.

***We agree and have removed this section, thank you for this suggestion.

Methods

- Most of the cohort is patients who do not have any ACSC. I think any unadjusted comparison between this group and the ACSC group (such as in figure 1) is fairly meaningless. They are clearly very different groups in terms of age etc.

***Agreed, we have removed this group from the figure.

- I didn't follow the rationale for the sample size. Surely it's a case of 'the more the better' – there was no need to restrict this sample to that of a largish CCG (300,000 patients) if more data were available.

***Indeed, more is always better! It is CPRD policy to limited datasets to 300,000, and the comparison to a CCG was made to put this number in context and show that it is large enough. We agree it is slightly confusing and have taken this comparison out.

- I agree that measuring potential access to care is difficult. However I have serious concerns around the proxy measure used by the authors due to potential confounding by underlying health, rurality and

a host of other factors that could feasibly be related to primary care utilisation and risk of hospital admission.

***Please see comment above

- The authors have attempted to adjust for this in one analysis by using matched controls, however I remain unconvinced by their matching strategy. Firstly, matching on the number of ACSCs may not be appropriate as some ACSC may impact the risk of admission much more than others (e.g. equating asthma with diabetes does not seem right). Secondly, I do not understand why they restricted their matching to only ACSC conditions. Other non-ACSC conditions are likely to influence the risk of admission (e.g. previous stroke) and could well differ between the hospitalised and non-hospitalised groups. Overall, I think the risk of bias in this matched case-control analysis is high and so its results cannot be fully trusted.

***The main reason for choosing to match based on the number of ACSCs, rather than specific diseases which would be more accurate, is that it would be difficult to find matches at this level of detail. Using the number of ACSCs, we were able to find a match for only 76% of the sample. It would be much lower if an exact match was required based on 10-year age band, gender, GP practice and specific disease profiles. We chose the number of ACSCs as they are most relevant to the analysis, and because we could count them and get an approximate 'degree' of ill health. We have added this justification to the discussion, as it is absolutely a limitation.

- I don't really see the relevance of the analysis exploring the impact of impact of primary care access on elective admissions, outpatient attendances etc., and the analysis exploring how primary care utilisation changes in the lead up to admission (given these are not mentioned in the aims)

***As per above, we agree and have removed this

Results

- An estimate of effect size (i.e. difference between cases and controls) should be given in the text and the results tables, currently only p-values are provided.

***Thank you for flagging this, effect size has been added

- The results for the patients with no ACSC are given in Figure 1 but not mentioned in the text. Really these patients offer very little to the study and could safely be removed from the paper entirely.

***As per above; removed

- The paper states that 'As observed for the long-term utilisation, there are factors such as the number of ACSCs that influence primary care utilisation'. I cannot see the data to support this statement. Can the authors clarify?

***We agree that this statement is confusing and incomplete, thank you for noting this. It was based on the fact that the difference between groups was made smaller by adjusting for confounding factors. We have removed the statement and rewritten the text to make it clearer.

Discussion

- I don't follow the rationale that because patients with an ACSC hospital admission were also more likely to have an outpatient procedure, this could have an impact on intervention design. Can the authors clarify? Are they saying that patients are more likely to be admitted as an emergency because they have had more outpatient appointments? This seems unlikely.

***We were thinking in terms of integrated care - if a patient also sees an outpatient specialist, coordination between primary care and outpatient care may have helped prevent complications. However, we have deleted the analysis on other settings as suggested, to keep the message of the paper clear. Therefore we have deleted this statement as well.

- The authors state that 'this paper only explores the relation between low access to primary care and

ACSC hospitalisation'. This does not appear to be true – for example they also examine outpatient utilisation

***As per above - has been deleted.

- The authors do acknowledge that confounding could drive their results however I think, given the limitations of the study, this should be stronger and explicitly mention that underlying health could be an important confounder which is not fully addressed in their case-control study.

***Fully agree, we have added this to the limitations and expanded on the problem of confounding and health status.

- I think that the conclusions stating that policymakers should focus on other aspects of primary care (e.g. continuity) are a little strong given the limitations of the study.

***Agreed, we have edited to ensure it says to focus on other aspects as well as access.

Reviewer: 4

Reviewer Name: Miguel Martín

Institution and Country: Facultad de Medicina, Unidad de Bioestadística, Universidad Autónoma de Barcelona

Please state any competing interests or state 'None declared': Non declared

Please leave your comments for the authors below

Although the described design is defined as a control case the analysis of the effect is performed by a linear model of repeated measures that does not clearly state what the variable response is.

***Thank you for noting this, we have added the response variable clearly to the design. As only one part of the analysis used a linear model of repeated measures, but all use the case-control, we used the latter to describe the study.

There are several aspects that I do not see clearly can facilitate the conclusion of the story.

Is the number of GP visits a measure of accessibility or severity of cases?

***In its unadjusted form, GP visits measure the severity of cases as well as accessibility. By using the case-control method, and adjusting for age and the number of ACSCs, we tried to correct for this so that only accessibility remains. We do agree that there are limitations to this, and have expanded our discussion on the topic of underlying health status.

Is accessibility or resolving capability being measured?

***We are unsure what is meant with 'measuring resolving capability'. If possible we would be happy to follow up with Dr Martin to discuss.

Why not carry out a follow-up analysis of the Poisson type, or related, with adjustment of the extravarience due to the appearance of the ACSC hospitalization interdependence? It is another approach but it provides the assessment of the risk of being hospitalized for the previous fact of "accessibility" and not so much the rate of hospitalization.

***The main reason we did not use a Poisson or other regression model to assess the risk of being hospitalised is that it would be difficult to determine the prior usage for 'no hospitalisation'. For people with ACSC hospitalisations, we can easily take the six months prior. However, we would also need to include people who do not have a hospitalisation in the model. But for this group its unclear which 6 month period to take. We solved this by matching each case to a similar control, and taking the same six months for the two.

Reviewer: 5

Reviewer Name: Gisele M Carriere, Senior Social Science Researcher

Institution and Country: Health Analysis Division, Statistics Canada, Canada

Please state any competing interests or state 'None declared': None declared.

Please leave your comments for the authors below

This is an important research question. A strength is the use of large population sample, especially since creation of nested case controls using exact matching of characteristics can challenge large enough sample sizes to look at relatively rare events like ACSC hospitalizations.

I answered 'no' to sufficient method description because the following things require some clarification: 1) how was the 'index' (my label) hospitalization selected for retrospective analysis of GP contacts? Very likely people with ACSCs had more than one hospitalization in this 5 year frame or even within 6 month frames. Perhaps several hospital events were chosen and the months before for each were compiled and pooled? Or maybe you removed people once they had a hospitalization within a given 6 month frame and considered them only once in that frame?

***Thank you for this comment, we did not explain this clearly in our paper. We included all ACSCs in the last 4.5 years of the 5 year study period (4.5 years to ensure all have a full six months of prior data available) - even if they were in the same person or if their '6 months prior' overlapped with another ACSC hospitalisation. As indeed many people have more than one hospitalisation for their ACSC, we did not want to exclude such a large proportion of ACSC hospitalisations. The retrospective analysis was done at the ACSC hospitalisation level rather than the patient level - so for every ACSC hospitalisation we identified the six months of GP utilisation prior to it. We have added this clarification to the paper.

2) Similarly, people may have 'accumulated' numbers of diagnosed ACSCs as time went on across the 5 years or even within a 6 month interval. How did you assign them one given # of ACSCs? What base year or month would have been used? I understand that 'time until hospitalization' was not of interest here (where a proportional hazards model would be used instead) -- rather this is a logistic approach. But I'm wondering how you considered changes that may have occurred for people to the #s of ACSCs over intervals used in the linear model.?

***We did not change the number of ACSCs over time, but used a cumulative number as measured over the 5 years. This is mainly because a code for COPD in year 2 of the study period does not signify that the disease first presented then - just that it received treatment at that point. With only 5 years of data, it is impossible to say whether a disease code at a certain point during the study is a newly onset disease, or treatment for an existing condition. We have clarified this in the methods.

It might be better to change language from "To explore changes in primary care " (implying trend or survival analysis) to 'To explore differences in primary care use leading up to ACSC hospitalizations...".

***Thank you for this suggestion, we have changed the language.

Next RE: Linkage - what method, or linkage approach was used to link primary care to secondary care in HES? What linkage rates resulted? Could you describe characteristics of those who could not be linked? i.e. perhaps they did not have GPs or other primary care providers?

***CPRD provided the primary care data for this research, and links it to HES data. Only GP registered patients who have a valid NHS identifier and have not opted out of data sharing can be linked to HES data. The patients are linked across the datasets based on NHS number, date of birth, sex and postcode (all information which is not available to us as researchers of course). Patients who are not registered with a GP practice are not covered by CPRD, and therefore not considered. We have added this information to the paper.

Is this a study of 'low access' (defined as ?) or does this examine association between primary and secondary care in general?

***The aim was to look at access, approximated using realised utilisation rates, because of the wide

belief that ACSC hospitalisations are a result of low access. It does explore an element of the association between primary and secondary care (i.e. what happens in secondary care as a result of lack of primary care) but there are many other interactions between the two that are not explored (e.g. referrals, discharge and readmission, A&E visits during evening hours when GP practices are closed, etc).

Minor: you have a typo in the last paragraph before the Discussion - 'significant different' should read 'significantly different'.

***Thank you! Edited

Lastly, Results are consistent to results by the following (2011) that showed individuals with ACSC hospitalizations were more likely to be users of primary and specialist care services and to have a regular doctor; risk factors for ACSC hospitalizations include such characteristics as having co-morbidities, low income and lifestyle some of which may be addressed by primary care however where other solutions to meet these health care needs my reside outside of the scope of health care: Sanmartin C, Khan S, Longitudinal Health Administrative Data Research team. Hospitalizations for ambulatory care sensitive conditions (ACSC): The factors that matter. Health Research Working Paper Series (Catalogue 82-622-X, No. 007) Ottawa: Statistics Canada, 2011. Thank you for the opportunity to review your research.

***Thank you for providing this interesting report, we have incorporated it in our paper.

VERSION 2 – REVIEW

REVIEWER	Leif Solberg HealthPartners Institute USA
REVIEW RETURNED	16-May-2017

GENERAL COMMENTS	The lack of a revision copy with tracked changes makes it somewhat difficult for me to be sure about what changes were made. Nevertheless, the authors appear to agree with my main point, although I don't see that the revision fully addresses my concern that case review would need to be added to reduce the limitations of reliance overmuch on big data for the conclusions made.
---

REVIEWER	William Brinson Weeks, MD, PhD, MBA The Dartmouth Institute for Health Policy and Clinical Practice Lebanon, NH USA
REVIEW RETURNED	17-May-2017

GENERAL COMMENTS	The authors have addressed all of my concerns. This is a relevant and interesting and thought provoking paper that examines a presumption about ACSC admissions in a thoughtful way.
--

REVIEWER	John Busby Queens University Belfast, UK
REVIEW RETURNED	05-Jun-2017

GENERAL COMMENTS	The authors have done a good job of addressing my concerns, however I still have lingering concerns around the validity of their
--

	analysis, and the opportunities for confounding to completely undermine their conclusions. However, these could not be easily addressed within the current study design. Apart from this there are a few minor points that could be easily fixed:  1. In the results section of the abstract I think the second paragraph should start 'In the short-term analysis' 2. It's a bit strong to say that that ACSCs should not require hospital care with high quality primary care. It would be more accurate to say that the amount of hospitalisations could be reduced. Similarly, I don't think it's fair to say that emergency ACSC admission can be considered as a failing in primary care (as some hospitalisations are completely unavoidable) – however it is probably the case that very high ACSC admission rates could be considered as a marker of poor quality primary care. The authors should probably tone the strength of their language down a bit here.
--	---

REVIEWER	Miguel Martin Universidad Autónoma de Barcelona Spain
REVIEW RETURNED	21-May-2017

GENERAL COMMENTS	Very interesting article and that continues leaving the door open to complementary studies. Perhaps the weakest point is to analyze the number of visits prior to hospitalization as a risk-increasing effect, instead of interpreting as an effect produced by the inefficiency of the system in solving problems or not considering the various levels of severity of the Problems studied. An additional point that is not clearly expressed is whether the authors also consider recurrent cases in the analysis, which would introduce a large increase in extravarience.
---

REVIEWER	Gisele M Carriere, Senior Social Science Researcher Health Analysis Division, Statistics Canada, Government of Canada Canada
REVIEW RETURNED	26-May-2017

GENERAL COMMENTS	Thank you for your fulsome responses to my earlier review comments. The revised manuscript is more clear. Possibly there remains need for concern about the fact that extreme cases were removed from the study (small counts for matched cases-controls) that could be stated more strongly but I am recommending ta the revised paper proceed to publication.
--

VERSION 2 – AUTHOR RESPONSE

Reviewer: 1

Reviewer Name: Leif Solberg

Institution and Country: HealthPartners Institute, USA

Please state any competing interests or state 'None declared': None declared

Please leave your comments for the authors below

The lack of a revision copy with tracked changes makes it somewhat difficult for me to be sure about what changes were made. Nevertheless, the authors appear to agree with my main point, although I don't see that the revision fully addresses my concern that case review would need to be added to reduce the limitations of reliance overmuch on big data for the conclusions made.

** We fully agree with the point made about needing case review to verify the results from administrative data, and have attempted to make this even more clear in our manuscript.

Reviewer: 2

Reviewer Name: William Brinson Weeks, MD, PhD, MBA

Institution and Country: The Dartmouth Institute for Health Policy and Clinical Practice, Lebanon, NH, USA

Please state any competing interests or state 'None declared': none declared

Please leave your comments for the authors below

The authors have addressed all of my concerns.

This is a relevant and interesting and thought provoking paper that examines a presumption about ACSC admissions in a thoughtful way.

** Thank you!

Reviewer: 4

Reviewer Name: miguel martin

Institution and Country: Universidad Autónoma de Barcelona, Spain

Please state any competing interests or state 'None declared': none declared

Please leave your comments for the authors below

Very interesting article and that continues leaving the door open to complementary studies.

Perhaps the weakest point is to analyze the number of visits prior to hospitalization as a risk-increasing effect, instead of interpreting as an effect produced by the inefficiency of the system in solving problems or not considering the various levels of severity of the Problems studied.

An additional point that is not clearly expressed is whether the authors also consider recurrent cases in the analysis, which would introduce a large increase in extravariance.

** We indeed include recurrent ACSC hospitalisations, and we have clarified this in the manuscript

Reviewer: 5

Reviewer Name: Gisele M Carriere, Senior Social Science Researcher

Institution and Country: Health Analysis Division, Statistics Canada, Government of Canada, Canada

Please state any competing interests or state 'None declared': None declared

Please leave your comments for the authors below

Thank you for your fulsome responses to my earlier review comments. The revised manuscript is more clear. Possibly there remains need for concern about the fact that extreme cases were removed from the study (small counts for matched cases-controls) that could be stated more strongly but I am recommending ta the revised paper proceed to publication.

** Thank you, we have added this limitation to the manuscript.

Reviewer: 3

Reviewer Name: John Busby

Institution and Country: Queens University Belfast, UK

Please state any competing interests or state 'None declared': None declared

Please leave your comments for the authors below

The authors have done a good job of addressing my concerns, however I still have lingering concerns around the validity of their analysis, and the opportunities for confounding to completely undermine their conclusions. However, these could not be easily addressed within the current study design.

Apart from this there are a few minor points that could be easily fixed:

1. In the results section of the abstract I think the second paragraph should start 'In the short-term analysis'

2. It's a bit strong to say that that ACSCs should not require hospital care with high quality primary care. It would be more accurate to say that the amount of hospitalisations could be reduced. Similarly, I don't think it's fair to say that emergency ACSC admission can be considered as a failing in primary care (as some hospitalisations are completely unavoidable) – however it is probably the case that very high ACSC admission rates could be considered as a marker of poor quality primary care. The authors should probably tone the strength of their language down a bit here.

** Thank you for these two points, we have addressed them both in the manuscript

VERSION 3 – REVIEW

REVIEWER	John Busby Queen's University Belfast, Northern Ireland
REVIEW RETURNED	23-Jun-2017

GENERAL COMMENTS	The authors have addressed my concerns
--

REVIEWER	Miguel Martín Unitat de Bioestadística Fac. Medicina Universitat Autònoma de Barcelona Spain
REVIEW RETURNED	29-Jun-2017

GENERAL COMMENTS	I believe that the authors' answers eliminate the doubts that I presented in my previous review. Regarding the use of acsc as an index of primary resolving capacity, perhaps that phrase is clear in the following articles, which unfortunately most of them are in Spanish, since our studies were fundamentally dedicated to the managers of the first level of Attention in my country, see for example: J. Caminal Homar, C. Casanova Matutano Aten Primaria 2003;31:61-5
--

Caminal J1, Starfield B, Sánchez E, Casanova C, Morales M.
Eur J Public Health. 2004 Sep;14(3):246-51.

Caminal J., Starfield B., Sánchez E., Hermosilla E., Martín M.
Primary Health Care and hospitalizations by Ambulatory Care
Sensitive Conditions in Catalonia. Revista Clínica Española, 2001,
201: 501-507.

In any case, congratulations